# Ethical Challenges in the Development of Virtual Assistants Powered by Large Language Models †

**Andrés Piñeiro-Martín** [1,2,*] , **Carmen García-Mateo** [2] , **Laura Docío-Fernández** [2] **and María del Carmen López-Pérez** [2]

1   Balidea Consulting & Programming S.L., Witland Building, Camiños da Vida Street, 15701 Santiago de Compostela, Spain
2   GTM Research Group, AtlanTTic Research Center, University of Vigo, Maxwell Street, 36310 Vigo, Spain; carmen.garcia@uvigo.es (C.G.-M.); ldocio@gts.uvigo.es (L.D.-F.); carmen.lopez@balidea.com (M.d.C.L.-P.)
*   Correspondence: andres.pineiro@balidea.com
†   This paper is an extended version of our paper published in IberSpeech 2022.

**Abstract:** Virtual assistants (VAs) have gained widespread popularity across a wide range of applications, and the integration of Large Language Models (LLMs), such as ChatGPT, has opened up new possibilities for developing even more sophisticated VAs. However, this integration poses new ethical issues and challenges that must be carefully considered, particularly as these systems are increasingly used in public services: transfer of personal data, decision-making transparency, potential biases, and privacy risks. This paper, an extension of the work presented at IberSPEECH 2022, analyzes the current regulatory framework for AI-based VAs in Europe and delves into ethical issues in depth, examining potential benefits and drawbacks of integrating LLMs with VAs. Based on the analysis, this paper argues that the development and use of VAs powered by LLMs should be guided by a set of ethical principles that prioritize transparency, fairness, and harm prevention. The paper presents specific guidelines for the ethical use and development of this technology, including recommendations for data privacy, bias mitigation, and user control. By implementing these guidelines, the potential benefits of VAs powered by LLMs can be fully realized while minimizing the risks of harm and ensuring that ethical considerations are at the forefront of the development process.

**Keywords:** ethical challenges; virtual assistants; Large Language Models; ethical AI; ethical guidelines; data privacy; bias mitigation; public services; AI regulation

## 1. Introduction

Artificial Intelligence (AI) has brought about significant advancements in many areas of our daily lives, and one of the most notable is the rise of AI-based Virtual Assistants (VAs). The appearance of Transformers [1] and effective transfer learning methods, such as ULMFiT [2] were the catalysts for models such as Generative Pretrained Transformer (GPT) [3] and Bidirectional Encoder Representations from Transformers (BERT) [4], which, together with unsupervised learning, have revolutionised almost every Natural Language Processing (NLP) benchmark. These breakthroughs have caused VAs to become increasingly intelligent, multilingual, and human-like in their interactions and they are being used in a growing number of contexts, including public services. Some projects of leading companies are truly remarkable and they open up new possibilities for developing more sophisticated VAs, such as Meta's CAIRaoke end-to-end neural model [5], Alexa's ability to recreate voices presented at Amazon's Re:Mars global AI event in 2022 [6], but mainly the arrival of ChatGPT on 30 November 2022 [7], with the Large Language Models (LLMs) as ChatGPT-3.5 first and as ChatGPT-4 [8] later, has been a clear turning point in terms of the popularity and development of integrated applications.

VAs have significant advantages, such as 24/7 service for real-time responses, cheap, easily replicable, multi-language, extensive knowledge capabilities, and available through

text or speech. They have the potential to respond to the growing demand for communication interfaces with the digital world, and the integration with LLMs opens up a world of possibilities unimaginable a few years ago. The standardisation of their use is leading appearance of conversational assistants in public services, such as health [9], education [10,11], and public administration. VAs powered by LLMs can help provide a better public service, unburden the overwhelmed system, save public funds, and increase the range of support and services available, allowing human resources to be dedicated to more useful and necessary services.

The increasing capabilities of the individual AI-based [12] components will enable conversations that are hardly distinguishable from real conversations with humans, but they will also raise concerns about bias, transparency, and accountability. The use of AI-based solutions needs to preserve the regulation and ethical principles, meaning that they must guarantee that the freedom, dignity, and autonomy of users are preserved, as well as that the rest of fundamental rights are respected. With such human-like performances and capabilities, especially in the public sector and when working with older populations, the risks and ethical implications of its use are huge [13]. One of the most significant risks is the potential for ageism and bias in the development and deployment of AI-based solutions. Older individuals may not be as familiar with new technologies, may have physical or cognitive impairments, or difficulty navigating complex user interfaces, which excludes them from the benefits of AI-based applications and reinforces negative stereotypes about aging and older people. Through our firsthand experience with NETA and GIDI, virtual assistants for e-health [14], we have witnessed the potential risks associated with AI-based solutions, particularly in the context of active aging or medical screening projects.

There is also the risk that AI-based virtual assistants may not respect people's privacy and confidentiality, particularly with regards to sensitive medical or personal information. The use of AI technologies in healthcare, social care, or education may require sharing data with third-party providers, which can compromise the people's privacy and autonomy. Moreover, these technologies may not adequately address issues related to safety and security, not detecting emergencies or providing incorrect or misleading information in certain situations, putting the safety and well-being of elderly individuals at risk.

Developers must be aware of these risks, meaning that there is potential for physical and psychological harm to humans, confusion, or bias in a service to which everyone contributes. There are many legal questions and ethical issues that arise when using these technologies: is it possible to explain how the system makes decisions? Can these data be transferred to third-party opaque AI services? Can the voice of a deceased relative be used? Should bots be humanized? Why are female voices used in most solutions? Should a bot react to racism or sexism? How do we prevent LLM hallucinations?

Answering these questions is a complex task [15,16]. There is still no consensus or clear regulation, and the use of LLMs is increasingly raising questions about their ethical and legal use. Italy has been the first country of the European Union (EU) to prohibit the use of ChatGPT, and by the time this paper is written, several countries are investigating its limitation and control (even in the United States, regulation is being debated, after Sam Altman, CEO of OpenAI, testified on 16 May 2023, before members of a Senate subcommittee and largely agreed with them on the need to regulate the increasingly powerful AI [17]).

There are many risks and fears that arise with powerful LLMs and their integration into VAs. The creation of credible phishing attempts, dissemination of inaccurate information, privacy concerns, biased content, plagiarism, and potential job displacement are some of the risks we have found in our VA projects that incorporate LLMs. Currently, the European Commission (EC) [18–20] (along with many other groups [21–23]) offers guidelines and assessments applied to AI with recommendations for the design of ethically acceptable solutions, and the EC is working on proposals to regulate the use of AI [24], however, none of them are expressly oriented to virtual assistants and the irruption of ChatGPT makes it probable that the European Commission's plan to regulate the use of IA requires revisions

or expansions. In order to deal with possible issues, it is necessary to legislate and regulate the use of AI. But even before that, to carry out an ethical analysis to establish more specific principles and guidelines in the case of virtual assistants and LLM-powered software.

The objective of this paper is, first, to analyse the current regulatory framework in Europe for AI-based virtual assistants and large language models, such as ChatGPT or PaLM 2 [25,26] for Google Bard. Based on this analysis and our work with voice-based conversational agents and their integration with LLMs, as a continuation of the work we presented at IberSPEECH 2022 [14], we aim to expose the legal and ethical issues that may arise when using this technology, particularly in public services. To do this, Section 2 analyses the ethics in design and the current regulatory framework for the development of AI-based solutions in Europe. Section 3 introduces our VAs and Section 4 outlines the risks and legal and ethical issues that arise when using VAs powered by LLMs. In Section 5, we present a discussion from an ethical standpoint and provide recommendations for the development and use of this type of technology. Finally, in Section 6, we present our conclusions and future directions.

## 2. Ethical Regulatory Framework

Ethics refers to the principles and values that govern human behavior, and encompasses the study of what is right and wrong, and how people ought to act in relation to others and the world around them. In the context of scientific research and development, ethical considerations are critical in ensuring that technology is designed and developed in a responsible and ethical manner. The Ethics by Design approach [27] incorporates ethical principles throughout the entire design and development process, allowing that ethical issues are addressed as early as possible and preventing negative impacts and risks to users and society. Based on the Charter of Fundamental Rights of the European Union (EU Charter), there are six general ethical principles that any AI system should preserve and protect:

1. Respect for human agency;
2. Privacy and data governance;
3. Fairness;
4. Individual, social and environmental well-being;
5. Transparency;
6. Accountability and oversight.

Currently, the European Union (EU) is one of the most restrictive places on earth in terms of data protection and the use of artificial intelligence. The European Commission (EC) offers guidance for adopting an ethically-focused approach while designing, developing, and deploying and/or using AI-based solutions [18,20], specifying tasks which must be undertaken. Although the proposals are for AI in general, it is possible to extrapolate them to the case of VAs and LLM powered applications. Proposed actions, in terms of VAs and LLMs, involve actions such as including external stakeholders in different phases of the project or assuming that any data gathered is biased, skewed, or incomplete until proven otherwise as part of the data collection.

Based on these guidelines, in 2020 the European Commission presented an assessment list for trustworthy artificial intelligence (ALTAI) for self assessment [19]. This list is made up of questions to help assess whether the AI system that is being developed, deployed, procured, or used, adheres to the seven requirements of Trustworthy Artificial Intelligence (the same six ethical principles of the EU Charter plus "Technical Robustness and Safety"). Although these assessments are much more specific than the guidelines, the action to be taken is not always clear. There are questions that directly impact VAs powered by LLMs in each requirement:

- Could the AI system generate confusion on whether a decision, content, advice or outcome is the result of an algorithmic decision?

Hallucinations are a well-known effect on the responses of Natural Language Generation (NLG) models and LLMs [28]. The quality and logic of the responses of VAs powered by LLMs could lead to misleading, incorrect, or fabricated data.

- Could the AI system affect human autonomy by generating over-reliance by end-users? Such human-like performance can lead to an overestimation of LLMs' capacities, causing overconfidence.
- Did you ensure a procedure to safely abort an operation when needed? VAs may include functionalities such as alerting the emergency services in exceptional cases.
- Did you implement the right to withdraw consent, to object, and to be forgotten? A VA could measure and store health and personal data, or transfer it to third-party LLMs services.
- Did you communicate the technical limitations and potential risks of the AI system to users, such as its level of accuracy and/or error rates? Being aware of the limitations and accuracy of a VA can avoid situations of frustration or risk.
- Did you try to include the participation of the widest range of possible stakeholders in the AI system's design and development? Large language models have been created by large companies in an opaque way in most cases, limiting stakeholder participation. By engaging relevant stakeholders, such as domain experts, ethicists, representatives from marginalized communities, and end-users, we can gather diverse perspectives and insights that can help address potential biases, enhance transparency, and promote accountability.

The European Commission is not alone in proposing guidelines and recommendations for the use of Artificial Intelligence. In 2018, the IEEE Global Initiative on Ethics of Autonomous and Intelligence Systems, composed of several hundred of participants from six continents, released the second version of the Ethically Aligned Design [29], where they provide insights and recommendations to ensure that every stakeholder involved in the design and development of autonomous and intelligent systems is educated, trained, and empowered to prioritise ethical considerations so that these technologies are advanced for the benefit of humanity. More recently, UNESCO has published Recommendations on the Ethics of Artificial Intelligence [30], addressed to Member States and paying specific attention to the central domains of UNESCO: education, science, culture, and communication and information.

However, adopting these guidelines, proposals and assessments remains voluntary in Europe, and, even with the release of ChatGPT, there does not seem to be a real demand from citizens for legislation on some of these issues [31] (since there is no collective awareness of the real possibilities or the risks that its use could entail) and, as mentioned in [32], "the industry lacks useful tools and incentives to translate high-level ethics principles to verifiable and actionable criteria for designing and deploying AI" [33].

*2.1. Regulating AI*

In April 2021, answering the regulation demand and the fact that many guidelines do not cover basic aspects of ethics [34], the European Commission presented a proposal to regulate AI [24]. This regulation will be the first in terms of AI and aims to implement an ecosystem of trust by proposing a legal framework for trustworthy AI. It is also based on EU values and fundamental rights and aims to give people and other users the confidence to embrace AI-based solutions, while encouraging businesses to develop them.

The regulation acknowledges the challenges posed by the opacity, complexity, biases, and partially autonomous behavior exhibited by certain AI systems. Even though striking the right balance in AI regulation is a complex endeavor, it requires ensuring safety and protecting fundamental rights without impeding innovation and hindering progress. Consequently, the proposal sparked extensive debate at the European level, resulting in numerous amendments being discussed to find the optimal balance between regulation and

avoiding excessive restrictions. This ongoing debate culminated on 11 May 2023, when the Internal Market Committee and the Civil Liberties Committee adopted a draft negotiating mandate with amendments to the Commission's proposal [35], which must be endorsed by the whole Parliament in June before negotiations with the Council on the final form of the law can begin.

This proposal is part of a package of measures the EC is working on to support the uptake of AI, which consists of three complementary strands: the legislative proposal mentioned above; a review of sectoral and horizontal rules on product safety; and EU rules to address liability issues related to AI systems. In relation to the last point, the proposal for a Directive on adapting non-contractual civil liability rules to AI was presented in September 2022 [36], and is intended to guarantee that people receive the same level of protection as in cases that do not involve AI systems (enhancing confidence in AI and promote its adoption within the EU). As mentioned in the proposal, safety and liability are two sides of the same coin: they apply at different moments and reinforce each other.

The regulation proposal follows a risk-based approach, which classifies AI systems into five categories with specific rules for each category:

- Prohibited AI systems;
- High-risk AI systems;
- Low-risk AI systems;
- Minimal-risk AI systems;
- General-purpose AI systems.

Most of the obligations apply to high-risk systems, those that may represent a risk to the health and safety, or to the fundamental rights of human beings. However, VAs and LLMs will not be considered high-risk as long as they do not conflict with any of the points in Annex III of [37]. If the system includes biometric identifications, such as Nuance's Gatekeeper service [38], integrated in some of their VAs, or AI-based health decisions, such as those described in [39], involving access to public services, they will be considered high-risk. When an AI is considered high-risk, it must meet a series of requirements, such as risk management system, data governance, technical documentation, transparency and provision of information to users, human oversight, accuracy, robustness, or cybersecurity.

For non-high-risk systems, the Commission will promote the voluntary application of the same requirements as for high-risks and the drawing up of codes of conduct for, for example, environmental sustainability, accessibility for persons with a disability, stakeholders' participation in the design and development of the AI systems and diversity of development teams on the basis of clear objectives and key performance indicators to measure the achievement of those objectives.

Finally, any AI system intended to interact with human beings, risky or not, will be in obligation to inform the user that he or she is interacting with an AI system (Article 52, Transparency obligations for certain AI systems [24]). Furthermore, the amendments to the regulation have established that generative foundation models (to which LLMs belong) should ensure transparency about the fact the content is generated by an AI system, not by humans, but that these requirements and obligations are not sufficient to consider these models as high-risk.

## 3. Virtual Assistants Powered by Large Language Models

In this section, we introduce NETA and GiDi, the virtual assistants for e-health in public services developed by Balidea. We also discuss the XIA project, which focuses on creating a virtual assistant for the electronic administration of the Galician government and incorporates LLMs. Lastly, we provide an overview of the tools utilized in the development of these projects.

The analysis of these projects and the ethical issues encountered have prompted us to examine the ethical principles that should be adhered to when designing a virtual assistant and how they should be implemented.

### 3.1. NETA and GiDi: VAs for e-Health

NETA (2018–2020) is a virtual assistant for active aging. This VA is integrated into a robot that speaks Galician and Spanish and aims to help the older people to remain in their environment as long as possible preventing hospitalization, promoting healthy lifestyle habits, supporting pharmacological and medical treatments, monitoring and providing security. In this way, the aim for the robot is to have a welfare task but also a social one, providing companionship through AI, facilitating video calls and encouraging active participation of the user in their community.

GiDi (2022–2024) is a project in collaboration with AXYN Robotique (https://www.axyn.fr/ (accessed on 18 July 2023)) for the creation of a virtual assistant for Screening Protocols at Home (SPH). GiDi will also help the older people to stay at home as long as possible preventing hospitalizations, it will facilitate the management of primary care and emergency services, avoiding saturations and always considering the social dimension of the patients. The project will be developed following an Ethics by Design approach, guaranteeing at all times the autonomy of the people and the explainability of the algorithms. The objective is to have experts in the field of medicine for the creation of screening protocols, as well as stakeholders involved in the project from the beginning. The Multimedia Technologies Group of the University of Vigo (https://atlanttic.uvigo.es/en/research/research-groups/gtm/(accessed on 18 July 2023)), the IDIS (Instituto de Investigación Sanitaria de Santiago de Compostela) (https://www.idisantiago.es/ (accessed on 18 July 2023)) Foundation and important companies in the social and public healthcare field are collaborating in this project.

### 3.2. XIA Project

The XIA project (2022–2024) is a collaborative research project between Balidea and the language technologies group of CITIUS (https://citius.gal/research/areas/ (accessed on 18 July 2023)) aimed at implementing conversational assistants to provide information for the Galician government's electronic administration procedures. These assistants must understand and speak both Galician and Spanish, both through voice and text. The purpose of these assistants is to offer citizens an additional channel for inquiries, available 24 × 7, with real-time responses for procedures related to accessing economic aids or obtaining official accreditations (in our case, the recognition of the degree of disability). The project has also followed an Ethics by Design approach, involving ethics committees that evaluated the entire project on two occasions. It also involved a legal department and collaborated with the administration and end users through regular meetings and focus groups.

In the first phase, the conversational assistants were similar to NETA and GiDi, where the information to be provided was modeled, and answers were generated based on templates or simple logic. In the second phase of implementation, we are exploring the integration with LLMs. This integration allows natural language responses to be generated based on the information provided as context to the LLM (the details of the integration are explained in Section 3.3). This integration significantly enhances the capabilities and domain of the conversational assistant.

### 3.3. Tools for the Creation of Virtual Assistants

Rasa [40], a set of tools that are open source python libraries for conversational software, is used for the development of the text-based conversational solutions. Rasa integrates the latest advances in NLP and AI, such as the Transformers [41,42] architectures. For the Speech to Text (STT) module, transfer learning and fine tuning of the XLS-R models [43] is applied using audios from Librispeech [44], from Common voice [45] and audios from an own corpus in Galician and Spanish. Finally, for the Text to Speech (TTS) module, the open source tool COTOVIA [46,47] is used, which supports Galician and Spanish.

For the second phase of the XIA project, we have utilized LangChain (https://python.langchain.com/docs/get_started/introduction.html (accessed on 18 July 2023)), a frame-

work for developing applications powered by language models, and integrated it with an OpenAssistant model [48] that we run on our local servers. In this type of architecture, the LLM is the central element, and through Rasa and NLP techniques we are able to analyse the user's message to identify a relevant context within the available documentation to provide the LLM. The LLM, based on the context provided and the user's question, generates the assistant's response.

## 4. Risk, Fears and Ethical Issues of VAs Powered by LLMs

As the capabilities of the LLMs continue to advance, so do the risks and ethical considerations associated with their application in virtual assistants. While LLM-powered virtual assistants offer numerous benefits, such as enhanced natural language understanding and improved user experiences, as we continue to explore their potential, it is essential to recognize and address the risks, fears, and ethical issues associated with their implementation.

Our understanding of these risks stems from our experience in developing virtual assistants and their integration with LLMs, from years of work in NLP, and from our firsthand experience and industry knowledge in projects, such as NETA, GiDi, and XIA. We are witnessing the immense benefits that LLMs technology brings, but we are also encountering and mitigating several challenges along the way, as a result of the analysis of the performance of our projects. These challenges have provided valuable insights into the potential risks and ethical considerations that must be carefully navigated to ensure the responsible and ethical deployment of LLM-powered virtual assistants.

**Privacy and Data Security**: One of the primary concerns associated with virtual assistants powered by LLMs is the collection and storage of user data. These assistants heavily rely on data input from users to improve their performance, which raises questions about data privacy and security. There is a fear that personal and sensitive information shared with virtual assistants could be vulnerable to unauthorized access or misuse, leading to potential privacy breaches and identity theft. The risk is especially high for VAs for public health or public administration services, which deal with critical personal data (administration information, health data, or education data). In our GIDI and XIA projects, we have observed that even though users are not authenticated, they often provide sensitive, personal, or confidential information during interactions with the assistants. This includes details such as their full name, income, medical conditions, identifiers, and more. Is it ethical to use external, non-auditable AI services (even if they are GDPR compliant)? Should AI systems in public services have full data governance?

**Bias and Discrimination**: LLMs learn from vast amounts of training data, which may inadvertently contain biases present in the data sources. Consequently, LLM-powered virtual assistants can inadvertently perpetuate or amplify biases in their responses, leading to discriminatory outcomes. The solution must guarantee that there will be no differences based on group, gender, religion, abilities, age, etc. How do we consider diversity (in the broad sense of the word)? How do we test our algorithms against bias? When working for public services, how do we involve public professionals and stakeholders? In our VA projects, we have actively involved stakeholders by incorporating them into the planning process and establishing collaborative agreements.

**Misinformation and hallucinations**: The ability of LLM-powered virtual assistants to generate coherent and contextually relevant responses also poses risks in terms of misinformation, especially with the presence of hallucinations, where the LLM generates text that goes beyond the scope of the provided input or fabricates information that is factually incorrect. These hallucinations raise important questions about the inner workings of LLMs and the underlying biases, training data, and algorithms that influence their language generation process. Understanding and mitigating the occurrence of hallucinations in LLMs is crucial to ensure the reliability, accuracy, and ethical use of these powerful language models, paving the way for their responsible deployment across a wide range of applications. When integrating this technology into public services, ensuring the accuracy and reliability of the information provided is paramount. During our tests of the LLM

integration in XIA, we observed instances where the model provided incorrect answers for accessing economic aids when the context was not appropriate. How can we distinguish between LLM hallucinations and reality? How can we develop tools to filter or detect hallucinations in LLMs? What is the ethical responsibility of LLM developers in relation to the hallucinations they may cause? Can we even use a system that hallucinates for public services?

**Dependency and Autonomy**: As virtual assistants become more integrated into our daily lives, concerns arise regarding their potential impact on human autonomy and decision-making. Excessive reliance on virtual assistants for tasks such as critical decision-making or personal organization may diminish individuals' sense of personal agency as they become increasingly dependent on technology. The performance and knowledge of these models are so remarkable that we have observed their ability to offer a higher level of security compared to human interaction, which may inadvertently contribute to an increased dependency on these systems. However, it is important to note that this exceptional performance can sometimes create a false sense of security. How can we strike a balance between the convenience offered by virtual assistants and the preservation of human autonomy? When incorporating virtual assistants into public services, are we prepared for them to serve as the sole source of information?

**Transparency and Explainability**: LLM-powered virtual assistants often operate as black boxes, making it challenging to understand their decision-making processes. Lack of transparency and explainability can raise ethical concerns, as users may not fully comprehend how their data are being used or how the assistant arrived at a particular response. Furthermore, although the logic for triggering a particular action is not usually AI-based in VAs (and if it were and would pose a risk to the user, the system would be labelled as high-risk and would be forced to comply with more restrictive rules), it is AI-based in the interpretation of the message and the generation of the response. How do we explain the algorithm's decisions when the inference is the result of such a complex process (or more complex than a particular prediction or classification process)? Is it necessary to explain the algorithm's decisions when these are not the rules that directly decide the actions to be taken by the algorithm?

**Training data and copyright**: It is well known that some of the data used to train LLMs consists of copyrighted texts, leading to concerns about potential infringement and legal complications. This raises several key challenges, including identifying ownership and licensing, navigating fair use and transformative use, and ensuring ethical data pre-processing and filtering. How can copyright ownership of data used to train LLMs be accurately determined, especially in cases of multi-authored datasets or complex data sources? What is the scope of fair use and transformative use in relation to the use of copyrighted data in LLMs, and how can they be appropriately applied in this context? Is it possible to address mutually beneficial solutions for developers and content creators?

**Inappropriate language**: For a VA to understand inappropriate language, they must be trained with it. Should the VA understand insults or inappropriate language? Should it react to this language? On the other hand, there are terms in decline with negative connotations but which are still used. For example, in Galician and Spanish, the word *"minusválido"* (handicapped) is a term in decline and clearly rejected by groups due to its pejorative connotation, however, it is still widely used. Should we include these terms in the training corpus?

**Name, voice, and gender of the assistant**: VAs tend to be female, at least in their default versions. Although there may be practical reasons for this (female voices seem to be more understandable and it is a well-established phenomenon that the human brain is developed to like female voices [49,50]), there are also gender stereotypes and assumed cultural biases [51,52]. How voice assistants may be designed to not perpetuate gender bias while promoting user adoption?

**VAs humanization**: The inclination to humanize bots as much as possible is evident: employing completely human voices, striving for increasingly natural interactions, simulat-

ing human behavior and emotions, and even incorporating jokes or casual conversations. The content generated by LLMs is now virtually indistinguishable from human-generated content. Nevertheless, there are potential risks when individuals lose awareness that they are interacting with a virtual assistant, particularly among older users who may easily forget the nature of the bot. This is a debate that we have encountered in our projects with the administration and public health services, particularly due to the significant interaction from older individuals. There is a genuine concern that these virtual assistants may be mistaken for humans, leading to potential frustration or excessive reliance on their responses. Should we continue humanizing the assistant, despite clearly informing users that it is an AI system? How can we prevent confusion and ensure a clear understanding of the virtual assistant's true identity?

**Stakeholders**: Stakeholders play a crucial role in the development of LLM-powered virtual assistants. Their input and involvement are essential for ensuring that the virtual assistants meet the needs and expectations of various user groups. Their engagement helps address ethical considerations, mitigate biases, enhance user experience, and ensure the responsible and inclusive deployment of LLM-powered virtual assistants in various domains. When working for public services, collaboration between stakeholders and public professionals is a must. Not just to obtain a good product, but to make sure that the solution will never pose a risk to the user and is fully validated by specialists (medical specialists, nurses, educators, teachers, or administrative employees). How do we involve them and ensure collaboration? At what stages is their collaboration necessary? How do we plan a dynamic process that ensures collaboration throughout projects' lifetime?

## 5. Discussion and Recommendations

This section presents a discussion on how to adapt general ethical standards to the specific case of Virtual Assistants powered by LLMs, and what rules or guidelines are needed where European regulation is scarce or non-existent.

### 5.1. Motivation

As discussed in Section 2, in accordance with the proposed regulation by the European Commission, it can be argued that a virtual assistant will not be classified as high-risk if it adheres to the criteria outlined in Annex III [37]. In the context of LLM-powered virtual assistants, they cannot be considered high-risk systems simply because they understand and generate responses equal to or superior to those of humans, and will only be considered high-risk when they include modules that make high-risk decisions. Some examples of high-risk virtual assistants include those utilized for assessing students in educational institutions, those incorporating some form of biometric identification, those granting access to essential private services or public services, or those enabling law enforcement authorities to perform assessments of individuals to evaluate risks, detect emotional states, or identify deep fakes.

However, as discussed in Section 4, there are many ethical issues that arise when using these systems, and their decisions can have a serious indirect impact on the user if these risks are not taken into consideration. For virtual assistants powered by LLMs that are considered non-high-risk systems, the only applicable regulations will be to always inform the user that he/she is conversing with an AI-based system and ensure transparency about the fact the content is generated by an AI system. In addition, the guidelines and recommendations in the regulations, while accurate, may be too general and broad for practical application. This is why we believe it is necessary to adapt and create more specific guidelines for their design, development, implementation and use of these systems, which although not considered high-risk, must be designed and used from an ethical point of view. On the other hand, as also pointed out in Section 4, some of the issues that arise when designing this type of solutions do not seem to have a clear answer in the guidelines and regulatory proposals, so we believe it is necessary to create new recommendations or rules to address them.

*5.2. Recommendations*

Out of our experience with virtual assistants powered by LLMs and on the analysis of the proposed regulations, a set of recommendations are presented below indicating the priority for their implementation:

**Auditability**: An essential requirement to comply with any of the recommendations in this paper is the auditability of the system by a third party. Without auditability, there is no guarantee that the system adheres to ethics by design. Firstly, auditability ensures transparency and accountability in the decision-making process of AI systems. By incorporating auditability, stakeholders can trace and understand the sources of biases, enabling the identification and mitigation of potential discriminatory or misleading outputs. Secondly, auditability enables the identification and rectification of errors or unintended consequences. LLMs are highly complex models, and there is a possibility of generating erroneous or misleading information. Auditing the outputs of LLMs provides an opportunity to detect such errors, enabling timely corrections and improvements to enhance the overall reliability and accuracy of the system. Furthermore, auditability contributes to advancing the field of AI. Thorough audits of LLMs provide researchers and developers with valuable insights into the model's behavior, strengths, and limitations. To achieve this, we propose that AI-based components of a public service assistant, along with high-risk systems, should be mandatorily registered in a public registry. Additionally, we firmly believe that LLMs, particularly due to their immense potential and associated costs, must be transparent and auditable systems. Utilizing opaque models in public services is not a viable option. Therefore, the public register should specifically encompass LLMs.

**Ensure privacy and data protection**: LLM-powered virtual assistants must adhere to data privacy regulations and follow best practices. This entails safeguarding users' personal and sensitive information, implementing data retention limitations, and offering transparent choices for informed consent and data control. When deployed in public services, it is imperative to ensure comprehensive data governance, particularly when handling personal, health, or education information. We firmly believe that the development of such assistants should be guided by legal and technical experts to ensure adherence to regulations. Given the complexity of this field, specialized advice is indispensable. In our view, utilizing services that do not provide such guarantees is not viable, regardless of the potential benefits and quality they may offer.

**Inclusion, diversity, and stakeholder collaboration**: It is crucial to promote inclusivity and diversity in the training data used for LLM. This entails considering diverse data sources and avoiding the exclusion of underrepresented groups to ensure fair and representative model responses. It is also essential to establish a collaborative channel between stakeholders and developers and ensure their involvement in the definition, modeling, and testing processes. When it comes to public service solutions, it is necessary to ensure a minimum level of collaboration among public professionals, stakeholders, and developers to achieve a valid product. Additionally, we consider it important to involve external ethics committees in the project analysis at various stages.

**Bias identification and mitigation**: LLMs are influenced by biases present in the training data. Techniques and processes need to be implemented to identify and mitigate biases related to gender, race, religion, and other factors that may impact model responses. As specified in the regulatory proposals, it should be assumed that any available resource is biased and, therefore, should be analysed. It is recommended to diversify data sources, conduct extensive data analysis, and apply data processing techniques to detect biases. Additionally, it is important to increase the diversity of development teams, implement rigorous testing and evaluation, where the model is analyzed in different scenarios, and establish feedback and transparency mechanisms to encourage users and stakeholders to identify biases. All this extracted information can be used at inference time by providing contexts to the LLMs or by post-processing the answers (if we know where it fails, we can try to avoid it). OpenAssistants publishes the guidelines that we should provide to the models to ensure a correct functioning (https://projects.laion.ai/Open-Assistant/docs/

guides/guidelines (accessed on 18 July 2023)), and these can be extended and improved thanks to the detected problems.

**Avoid confusion and informed consent**: Because conversations may be indistinguishable from those with a human, users must be aware at all times that they are interacting with an AI-based system. For this, in addition to informing, it is recommended to avoid incorporating human behaviors, emotions or jokes in the VA responses as well as to detect possible confusions in order to act and insist on their nature, especially when the system is designed to work with older people. Furthermore, obtaining informed consent from users is essential for collecting and using their data. Users must have clarity on how their data will be used and have the option to opt out if desired.

**Continuous Evaluation**: AI-based VAs and LLMs are evolving products, in need of improvements and adjustments, that greatly benefit from real usage data. VAs powered by LLMs should undergo regular evaluations to identify potential ethical issues and enhance their performance. This entails monitoring user interactions, gathering feedback, and being willing to make adjustments and improvements based on the findings. There is also the need to extend and ensure collaboration between developers and stakeholders throughout throughout projects' life, in order to monitor, detect ethical issues, and validate updates and new developments. We believe that collaboration contracts that do not provide for such an extension of the project are not admissible.

**Gender**: Avoid gender role assumptions when selecting the voices and the name of assistants (e.g., nurses or assistants with female names and voices, doctors or security information with male names and voices), and give the option to select the gender of the voice. It is also recommended to include non-gendered names and voices (if available).

**Terms in decline and inappropriate language**: It is recommended to include it in the corpus in order to at least detect it and, if desired, act accordingly. In this way, the VA can have a pedagogical role, informing about the accepted terms, notifying in case of detection, or redirecting the conversation.

**Publicly available data**: This is one of the most debated issues, and we think that is highly dependent on the nature and context of AI systems. Releasing training data promotes transparency and explainability, allowing researchers, experts, and the wider community to examine how models have been trained in line with accountability. It also has positive implications for reproducibility, verifiability, and research advancement. However, it also raises legitimate concerns about the privacy and confidentiality of data. In our opinion, it is highly advisable to release models training data. If not completely unrestricted, it should be made available as part of a registry accessible to regulatory authorities or public services. This has been the approach we have adopted for the FalAI dataset, which is a voice dataset for End-to-End (E2E) Spoken Language Understanding (SLU) collected by Balidea and the University of Vigo [53]. The dataset, in line with the commitment to diversity, involves the participation of thousands of individuals spanning different age groups. It is currently the largest publicly accessible dataset of its kind and adheres to the practices embraced by the research community in Europe. As part of this, we have published the initial version of the dataset (Accessible via: https://live.european-language-grid.eu/catalogue/corpus/21575 (accessed on 18 July 2023). As soon as the validation process is complete, all data will be published freely) on the European Language Grid (ELG) [54], a high-quality repository with multiple licensing and data release alternatives.

Releasing data under controlled frameworks can facilitate compliance with privacy regulations. It offers a solution that enables companies to share their training data while preserving their competitive advantage and mitigating concerns of exploitation by competitors. By implementing controlled access to the data, companies can strike a balance between promoting transparency and safeguarding their proprietary knowledge. This approach allows regulators and relevant stakeholders to assess fairness, ethical considerations, and potential biases within the training data without exposing it to unauthorized parties. In summary, by releasing training data under carefully designed frameworks, limited to authorized entities such as regulators, companies can strike a balance between promoting

transparency and protecting their market position. This approach facilitates collaborative evaluation of AI systems while mitigating risks associated with data exploitation and preserving proprietary knowledge.

*5.3. Limitations*

While the recommendations provided in Section 5.2 offer valuable insights for the implementation of ethical guidelines in LLM-powered virtual assistants, it is important to recognize that there are certain considerations and factors to be aware of:

**Applicability limitations**: The recommendations presented in this paper may have limitations in terms of their applicability to different contexts and AI systems. The effectiveness and feasibility of these recommendations may vary depending on factors such as the specific use case, available resources, regulatory frameworks, technological capabilities, and domain application. It is essential for stakeholders to carefully consider the applicability of each recommendation in their respective contexts and adapt them accordingly.

**Regulatory constraints**: The proposed recommendations are based on an analysis of the current regulatory landscape and proposed regulations by the European Commission. However, regulatory frameworks and requirements may evolve over time, potentially impacting the applicability and relevance of these recommendations. It is crucial to stay updated with the latest regulations and ensure compliance with the specific legal requirements in each jurisdiction.

**Technical challenges**: Implementing certain recommendations, such as auditability and bias identification, may pose technical challenges. Developing robust auditing mechanisms, detecting and mitigating biases effectively, and ensuring privacy and data protection can require significant technical expertise and resources. Organizations may encounter difficulties in implementing these recommendations due to limitations in technology, infrastructure, or expertise.

**Ethical dilemmas**: The implementation of certain recommendations may give rise to ethical dilemmas and trade-offs. For example, releasing training data for transparency purposes can raise concerns about privacy and the potential misuse of data. Balancing transparency with the need to protect sensitive information and proprietary knowledge poses ethical challenges that require careful consideration.

**Limitations of the research**: This study has its limitations in terms of scope and depth. The recommendations presented are based on the analysis of existing literature, regulations, and the expertise of the authors, drawing from their experience working on various projects related to virtual assistants; however, it is important to acknowledge that there may be other perspectives and factors that were not fully considered in this research. Future studies and ongoing research in the field of AI ethics can further refine and expand upon these recommendations.

It is crucial to approach the implementation of these recommendations with a critical mindset, considering the specific context and limitations of each situation. Stakeholders should engage in continuous evaluation and improvement to address emerging challenges and adapt the recommendations accordingly.

## 6. Conclusions

In this paper, we have presented a set of recommendations for the utilization of virtual assistants powered by large language models, aiming to enhance and complement the existing European framework. To accomplish this, we have examined prevailing guidelines, assessments, and regulatory proposals concerning AI usage in Europe. Furthermore, we have deliberated on adapting these recommendations to address the challenges we encountered in our practical experience with virtual assistants powered by LLMs. In cases where existing guidelines or proposals lacked clarity, we have proposed new guidelines that align with the existing ones.

The recommendations provided in the article cover several key aspects. These encompass advocating for the auditability of the systems by declaring them in public registers,

seeking guidance and expertise from privacy and data protection professionals, establishing effective channels of collaboration among project stakeholders, professionals, and users, implementing robust feedback and transparency mechanisms to identify and address biases, clearly informing users about the nature and limitations of the assistant, and releasing training data under controlled frameworks. Given the rapid advancements in LLMs, it is crucial to continuously adjust the suggested regulations while tailoring existing ones to specific applications.

However, addressing these risks, concerns, and ethical issues necessitates a comprehensive approach involving developers, regulators, and society as a whole. The implementation of privacy safeguards, regular bias audits, promotion of algorithmic transparency, and empowering users with control over their data are essential measures in fostering responsible and trustworthy LLM-powered virtual assistants. Additionally, ongoing dialogue and collaboration among stakeholders will be pivotal in navigating the intricate landscape of ethical considerations and ensuring the ethical deployment of LLM technology in virtual assistants.

*Future Directions*

As mentioned in Section 2.1, the ongoing examination of regulatory proposals in Europe will lead to extensive discussions on the proposed amendments in the coming months. These amendments aim to establish the world's first regulation of its kind. The final shape of this ambitious project's legislation will require careful scrutiny, and we anticipate that future research should focus on studying and tailoring the European regulation to specific technologies or use cases, such as virtual assistants powered by LLMs.

Furthermore, exploring the potential inclusion of provisions addressing the interpretability and explainability of LLM-powered systems within the regulatory framework could be a promising path for future investigation. This would contribute to enhancing transparency and accountability, promoting user trust, and ensuring the ethical deployment of LLM technology in various applications.

**Author Contributions:** Conceptualization, formal analysis, investigation and writing—original draft, A.P.-M.; Writing—review and editing, A.P.-M., C.G.-M., L.D.-F. and M.d.C.L.-P. All authors have read and agreed to the published version of the manuscript.

**Funding:** This work is funded by the Galician Innovation Agency (GAIN) and the Consellería de Cultura, Educación, Formación profesional e Universidades of the Xunta de Galicia through the program: Doutoramento Industrial [55]. It has also received funding from the Consellería de Cultura, Educación, Formación profesional e Universidades of the Xunta de Galicia for the "Centro singular de investigación de Galicia" accreditation 2019–2022 and by the "Axudas para a consolidación e estructuración de unidades de investigación competitivas do Sistema Universitario de Galicia -ED431B 2021/24", and the European Union for the "European Regional Development Fund-ERDF".

**Data Availability Statement:** Not applicable.

**Conflicts of Interest:** The authors declare no conflict of interest. The funders had no role in the design of the study; in the collection, analyses, or interpretation of data; in the writing of the manuscript, or in the decision to publish the results.

## Abbreviations

The following abbreviations are used in this manuscript:

| | |
|---|---|
| AI | Artificial Intelligence |
| ALTAI | Assessment List for Trustworthy Artificial Intelligence |
| BERT | Bidirectional Encoder Representations from Transformers |
| CITIUS | Centro Singular de Investigación en Tecnoloxías Intelixentes |
| EC | European Commission |
| ELG | European Language Grid |

| | |
|---|---|
| EU | European Union |
| E2E | End-to-End |
| GAIN | Galician Innovation Agency |
| GPT | Generative Pretrained Transformer |
| IEEE | The Institute of Electrical and Electronics Engineers |
| LLM | Large Language Model |
| NLG | Natural Language Generation |
| NLP | Natural Language Processing |
| PaLM | Pathways Language Model |
| SLU | Spoken Language Understanding |
| STT | Speech To Text |
| TTS | Text To Speech |
| ULMFiT | Universal Language Model Fine-tuning |
| UNESCO | United Nations Educational, Scientific and Cultural Organization |
| VA | Virtual Assistant |
| XLS-R | Cross-Lingual Speech Representation |

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
