# Peer review of "Ethical Challenges in the Development of Virtual Assistants Powered by Large Language Models"

_electronics, doi:10.3390/electronics12143170_

Round 1

Reviewer 1 Report

The paper presents exploration of ethical considerations and recommendations for the design and application of large language model virtual assistants. The paper is well organized. Particularly valuable are the specific recommendations.

An innovative element is the advocacy for releasing training data under controlled frameworks, that can contribute to transparency and ethical compliance without violating privacy or commercial interests.

There are some areas in which the paper could be improved:

- The paper could benefit from more specific case studies or real-world examples to support the arguments.

- Some concepts could be explained for the non-specialist reader, e.g. what constitutes a 'high-risk decision' by a virtual assistant, with some examples of such decisions.

- The authors make a strong case for recommendations, but discussion on potential limitations and the challenges in implementing such recommendations could provide a more balanced view.

- The limitations of the research (threats to validity) could also be taken into account.

Overall, paper contributes significantly to the ongoing field around the ethical use of LLMs in virtual assistants. 

Reviewer 2 Report

A detailed review was attached.

Round 2

Reviewer 2 Report

Accept in present form